# Entrapment, Hopelessness, and Cognitive Control: A Moderated Mediation Model of Depression

**DOI:** 10.3390/healthcare11081065

**Published:** 2023-04-07

**Authors:** Hyunju Choi, HaeJin Shin

**Affiliations:** 1Department of Education, University of Ulsan, Ulsan 44610, Republic of Korea; 2College of Liberal Arts, Seoul National University of Science and Technology, Seoul 01811, Republic of Korea

**Keywords:** entrapment, hopelessness, depression, cognitive control, moderated mediation model

## Abstract

The present study aimed to examine the roles that hopelessness and cognitive control play in the relationship between entrapment and depression. Data were collected from 367 college students in South Korea. The participants completed a questionnaire that consisted of the Entrapment Scale, the Center for Epidemiologic Studies Depression Scale, the Beck Hopelessness Inventory, and the Cognitive Flexibility Inventory. Results showed that hopelessness partially mediated the relationship between entrapment and depression. In addition, cognitive control moderated the relationship between entrapment and hopelessness: greater cognitive control weakened the positive association between entrapment and hopelessness. Finally, the mediating effect of hopelessness was moderated by cognitive control. The findings of this study expand the understanding of the protective role of cognitive control, especially when an increased sense of entrapment and hopelessness intensifies depression.

## 1. Introduction

Studies have shown that entrapment and hopelessness are robust predictors of depression [1,2]. Although both variables are cognitive perceptions of adversity, hopelessness differs from entrapment in that it refers to a way of thinking about the future. Hopelessness is characterized by a relative lack of motivation to escape from distressing situations, and being pessimistic about the future [3]. In contrast, entrapment refers to the feeling of being currently trapped in a stressful situation, despite having a strong desire to escape [1,4,5]. Researchers suggest that entrapment may develop into depressive symptoms via hopelessness [6,7]. For example, the Cry of Pain model theorized a suicidal pathway whereby entrapment develops into hopelessness, which in turn triggers suicidal behaviors. The model pinpoints that hopelessness is associated with one’s perceptions, especially when frustrated escape is projected toward the future [6,7]. Given the close link between depression and suicidality, it may be also plausible that hopelessness mediates the relationship between entrapment and depression. Few studies, however, have examined how entrapment and hopelessness interplay in aggravating depression.

Achieving a sense of control is one of the most effective ways to alleviate entrapment. Individuals who feel trapped often believe that they have no control over their circumstances [8]. Similarly, according to the helpless model of depression, repeated failures to cope with stressors intensify one’s lack of control over adverse events. Such helplessness dampens the sense of hope, leading to depression [9]. Given all this, cognitive control, which refers to one’s perception that challenging situations are still controllable [10], may moderate the negative effects of both entrapment and hopelessness against depression. The present study aimed to investigate the associations among entrapment, cognitive control, hopelessness, and depression by testing a moderated mediation model.

### 1.1. The Mediating Role of Hopelessness

Entrapment is defined as a defensive process. Individuals who feel trapped wish to escape from stressful situations but perceive them to be inescapable [8]. Central to the concept of entrapment is subjective emotions rather than objective evaluations of circumstances [3]. According to the arrested defenses model, depression can be triggered by a sense of entrapment [8]. When a person encounters an adverse situation, he or she may use either flight or fight strategies to cope with stressors. However, such defenses are generally effective only for a short period of time. If these coping responses are activated but, subsequently, blocked or arrested, then an individual is more likely to experience chronic stress despite the strong motivation to escape. Several studies have shown that there are positive links between entrapment and depression [11,12]. For example, Taylor and his colleagues [3] reviewed fifty-one studies that examined the relationships between entrapment and depression, using clinical and nonclinical samples, and found that the overall effect was moderate in magnitude (*r* = 0.56).

While entrapment has a direct effect on depression, hopelessness may serve as a pathway through which entrapment indirectly affects the symptoms of depression. Hopelessness has also been found to be a major predictor of depression [9,13]. In general, hopelessness refers to pessimistic views about the future [13]. Both hopelessness and entrapment are experienced due to failures to solve problems [5]. However, entrapment, which indicates beliefs that one’s current circumstances are unchangeable and uncontrollable, often evolves into a sense of hopelessness [4,14].

According to the hopelessness theory of depression, individuals who consistently fail to cope with aversive stressors are likely to believe that their current circumstances are inescapable. Such a sense of entrapment may intensify their belief that their future also affords limited opportunities for positive change. This sense of hopelessness eventually leads to depression [15]. Indeed, a similar pathway has been empirically investigated in relation to suicidal behaviors. Littlewood and her colleagues [7] found that hopelessness mediates the relationship between entrapment and suicidal behaviors. However, to the best of our knowledge, no past study has examined the mediating effect of hopelessness on the relationship between entrapment and depression. Accordingly, the present study aimed to bridge this gap in the literature by examining the indirect effect of entrapment on depression through hopelessness among college students.

### 1.2. The Moderating Effect of Cognitive Control

Cognitive control is a subfactor of cognitive flexibility, which refers to the ability to switch between different cognitive sets in order to successfully adapt to changing circumstances [10]. Researchers have found that perceived control is negatively associated with hopelessness. For instance, an external locus of control, which refers to the belief that the results of one’s behaviors are determined by chance or controlled by powerful others, has been found to be associated with diminished hope [16]. These findings suggest that a sense of control may serve as a buffer against the effects of entrapment on hopelessness. Furthermore, a lack of cognitive control is a significant predictor of psychological problems such as depression and anxiety [17,18]. For example, Junco and his colleagues [18] found that cognitive control was positively associated with reduced subjective emotional distress by promoting cognitive restructuring. Other studies have also shown that cognitive control is negatively related to the use of maladaptive emotion regulation strategies [19,20].

These findings suggest that perceived control plays a risk-buffering role in the association between entrapment and its undesirable consequences. To be specific, if hopelessness mediates the relationship between entrapment and depression, and cognitive control moderates the relationship between entrapment and hopelessness, then the mediating effect of hopelessness should be moderated by cognitive control as well.

### 1.3. Control Variable

Previous research [21] has revealed gender differences in entrapment, cognitive control, hopelessness and depression. For example, Shin (2019), in her research examining gender differences in the moderation effect of cognitive control on depression, found that gender differences existed regarding the moderating effects of cognitive control on entrapment [21]. To be specific, internal entrapment moderated the relationship between female students’ perceived stress and depression, whereas external entrapment moderated the relationship between perceived stress and anxiety among female students. Further, gender differences in hopelessness have been studied, with some research suggesting that women may be more likely than men to experience feelings of hopelessness [22]. Additionally, the research reports gender differences in depression due to the effects of genes and gender-specific manner. Such results indicate the need for examining and testing gender as a control variable. In addition, grade level and age are considered control variables since students’ experiences are more likely to be different according to their grade levels.

### 1.4. The Present Study

We predicted that college students with high levels of entrapment would be more likely to experience depression, and that the relationship between entrapment and depression is mediated by hopelessness. This hypothesis was formulated based on the premise that feelings of being currently trapped despite the strong desire to escape are positively associated with a pessimistic view of the future (feelings of hopelessness) [7]. Given that cognitive control serves as a buffer against the negative effects of entrapment, we also predicted that cognitive control moderates the negative effects of entrapment on hopelessness. Finally, the mediating effect of hopelessness is expected to vary across college students with different levels of cognitive control. In other words, college students with lower levels of cognitive control were expected to be more likely to report higher levels of hopelessness (which, in turn, were expected to predict higher levels of depression) than their counterparts with higher levels of cognitive control.

The proposed conceptual model is presented in Figure 1. Based on the literature review and our discussion on previous research results, we formulated three hypotheses. First, we hypothesized that hopelessness mediates the relationship between entrapment and depression. Second, we hypothesized that cognitive control moderates the relationship between entrapment and hopelessness. Lastly, we hypothesized that cognitive control moderates the mediating effect of hopelessness on the relationship between entrapment and depression.

## 2. Materials and Methods

### 2.1. Participants

Data were collected from college students in Seoul and the south-eastern regions of South Korea using convenience sampling. The participants were 367 (males: 178, females: 187, unknown: 2) students who volunteered to participate in this study as a part of a class exercise. The average age of the participants was 21.84 (standard deviation 2.16). The sample consisted of 50 (13.8%) freshmen, 124 (34.2%) sophomores, 95 (26.5%) juniors, and 94 (25.9%) seniors.

### 2.2. Measures

#### 2.2.1. Demographics

All participants answered questions concerning gender, grade level, and age. Participants were allowed to write in their gender (male or female). Grade response options were freshmen, sophomores, juniors, and seniors. ‘Decline to state’ options were also available. Gender, age, and grade level were included and tested as control variables.

#### 2.2.2. Entrapment

Entrapment was measured using the Internal Entrapment Scale (IES) and External Entrapment Scale (EES) [1]. Both of them assess feelings of being trapped despite the desire to escape. The Korean version of the Entrapment Scale was developed and validated by Lee and Cho [23]. It consists of two subscales. The EES consists of 10 items (e.g., “I would like to get away from other more powerful people in my life”). In contrast, the IES consists of 6 items, which assess the feeling of being trapped by one’s own internal thoughts (e.g., “I want to get away from myself”).

#### 2.2.3. Depression

Depression was assessed by adopting the Center for Epidemiologic Studies Depression Scale (CES-D) [24], which is one of the most widely used measures of depression. The Korean version of the CES-D has been translated into Korean and validated by Chon and his colleagues [25]. This scale consists of 20 items that assess the frequency with which the respondent has experienced the symptoms of depression during the past week. Responses are recorded on a scale that ranges from 1 (almost never) to 4 (almost always). Four items are negatively worded. Therefore, responses to these items were reverse-scored. Composite scores were computed by summing individual item scores. Higher scores indicate greater symptoms of depression. The Cronbach’s alpha of this scale was 0.89 in this study.

#### 2.2.4. Hopelessness

The Korean version of the Beck Hopelessness Scale (BHS) [26], which has been validated by Kim and his colleagues [27], was used to measure hopelessness. The BHS consists of 20 true-false items that assess positive and negative beliefs about the future. Each item is scored 0 (Hopeful) or 1 (Hopeless). The questionnaire includes items such as “I look forward to the future with hope and enthusiasm” and “I might as well give up because there is nothing that I can do about making things better for myself.” Nine items were reverse-scored and were summed for the total score. Total scores on the BHS range from 0 to 20, and higher scores are indicative of a higher level of hopelessness. In this study, the Cronbach’s alpha of this scale was 0.93.

#### 2.2.5. Cognitive Control

The Cognitive Flexibility Inventory (CFI) [10] was used to assess cognitive control. Heo [28] translated the CFI into Korean and validated it using Korean samples. The CFI is a 20-item self-report measure that assesses the extent to which an individual demonstrates flexibility in how he or she perceives his or her circumstances and copes with stressors. The measure consists of two subscales: cognitive alternatives (13 items) and cognitive control (7 items). In this study, only the cognitive-control subscale was used. Responses are rated on a scale that ranges from 1 (Never) to 7 (Very much so). Higher subscale scores are indicative of a greater sense of control over one’s circumstances. The following is a sample subscale item: “I am capable of overcoming difficulties in life”. The Cronbach’s alpha of this subscale was 0.78 in this study.

#### 2.2.6. Procedure

The researchers contacted faculty members who were teaching undergraduate courses at four-year universities in Seoul and the south-eastern regions of South Korea. With their permission, students were informed about the study and its purpose through in-class presentations. Participants were provided with informed consent forms and questionnaires. All the respondents were treated in accordance with the code of ethics and conduct of the Korean Psychological Society [29]. In total, 392 survey packets were distributed; 367 participants returned completed informed consent forms and questionnaires (response rate: 94%). Researchers’ contact information, including email addresses and phone numbers, was provided to the participants so that they could contact them regarding any further questions about the research process.

#### 2.2.7. Statistical Analysis

All analyses were conducted using the SPSS program. Descriptive statistics and correlation coefficients were computed to examine basic trends that underlay the data. Moderated mediation analysis was conducted using the PROCESS macro for SPSS [30]. The indirect effects and index of moderated mediation were estimated using bootstrapping and 5000 bootstrap samples, and 95% bootstrap confidence intervals were computed. The indirect effect and index of moderated mediation were considered to be statistically significant if the confidence interval did not include zero.

## 3. Results

### 3.1. Preliminary Analysis

Table 1 presents descriptive statistics for the study variables and intercorrelations among them. Bivariate correlation analysis was conducted to examine the relationships between the study variables. Entrapment was positively correlated with hopelessness (*r* = 0.28, *p* < 0.001) and depression (*r* = 0.77, *p* < 0.001), but negatively correlated with cognitive control (*r* = −0.62, *p* < 0.001). Cognitive control was negatively correlated with hopelessness (*r* = −0.27, *p* < 0.001) and depression (*r* = −0.58, *p* < 0.001). Hopelessness was positive related to depression (*r* = 0.32, *p* < 0.001). T-test analysis revealed that gender differences exist in entrapment (*t*= −6.59, *p* < 0.001), cognitive control (*t* = 5.44, *p* < 0.001), and depression (*t* = −5.98, *p* < 0.001). The scores of female students on entrapment and depression were higher than those of male students. In contrast, scores on cognitive control were significantly higher among male students than among female students. There were no statistically significant differences in terms of age or grade level.

### 3.2. Hypothesis Testing

Mediation analysis was conducted to test hypothesis 1. As shown in Table 2, the regression coefficients that emerged from the relationship between entrapment and hopelessness (B = 1.76, *p* < 0.001), as well as from the relationship between hopelessness and depression (B = 0.01, *p* < 0.01), were significant after controlling for the effect of age and sex. The bootstrapped indirect effect was 0.019 (bootstrap standard error = 0.01), and the 95% confidence interval ranged from 0.007 to 0.034. This indicated that the indirect effect of hopelessness was statistically significant. In other words, higher levels of entrapment were related to higher levels of hopelessness, which, in turn, were related to higher levels of depression. Thus, hypothesis 1 was supported.

To test hypotheses 2 and 3, a moderated mediation analysis was conducted. As predicted, cognitive control moderated the relationship between entrapment and hopelessness (Table 3). The interaction between entrapment and cognitive control had a significant effect on hopelessness (B = −0.83, *p* < 0.05). The result indicated that the effect of entrapment on hopelessness differed across students with varying levels of cognitive control. Figure 2 shows these interaction patterns. The positive relationship between entrapment and hopelessness was stronger among students with lower levels of cognitive control (low; β = 1.497, *p* < 0.01) than among those with average (medium; β = 0.829, *p* > 0.05) and higher (high; β = −0.005, *p* > 0.05) levels of cognitive control. Thus, hypothesis 2 was supported.

The results also support the conditional indirect effect of cognitive control. The index of moderated mediation was −0.009 (bootstrap standard error = 0.004, [−0.018, −0.002]). These results indicate that the mediating effect of hopelessness on the relationship between entrapment and depression differed across groups with varying levels of cognitive control. Among students with lower (M—1 SD) levels of cognitive control, hopelessness mediated the relationship between entrapment and depression (indirect effect = 0.016, [0.004, 0.032]). However, among students with average (M) and higher levels of cognitive control (M + 1 SD), the mediating effect of hopelessness was not statistically significant (indirect effect = 0.009, [−0.001, 0.022]; indirect effect = 0.000, [−0.014, 0.013]). Thus, hypothesis 3 was supported. The final model is presented in Figure 3.

## 4. Discussion

### 4.1. The Mediating Effect of Hopelessness

Consistent with past findings, entrapment had a significant positive effect on depression in the present study. According to the arrested defense model, entrapment can be depressogenic because it contributes to the dysregulation and exacerbation of flight or fight coping responses [11]. Flight or fight responses tend to be initially adaptive. However, the prolonged use of such coping strategies is likely to result in psychological dysfunction (e.g., negative self-referential cognitions, feeling out of control, and a lack of positive emotions), which, in turn, can exacerbate depression [31]. Central to the arrested defenses model is the conceptualization of entrapment as a subjective appraisal rather than an objective circumstance [1,3]. By highlighting cognitive appraisals of one’s circumstances as an important predictor of depression, the present findings provide additional empirical support for the validity of the arrested defenses model.

It has been reported that people with a high sense of entrapment are more likely to experience depression and anxiety [3,32]. Supporting the current result, previous studies suggest that feelings of entrapment significantly explain depression even after controlling for socioeconomic status-related variables and past history of depression [1]. In particular, Brown and colleagues (1995) believed that the feeling of entrapment or shame caused by the stressful experience explained depression better than the experience itself. For example, a loss that entails a sense of entrapment (e.g., separation due to infidelity, divorce) is three times more likely to cause depression than a loss that is not accompanied by a sense of entrapment or shame (e.g., death of a family member) [33]. The current result is further empirical evidence that entrapment plays a key role in predicting depression. It is also noteworthy that hopelessness partially mediated the relationship between entrapment and depression. Entrapment not only has a direct effect on depression, but it also has an indirect effect through hopelessness. Thus, the current finding sheds light on the psychological process of how a sense of hopelessness bridges the link between entrapment and depression. When an individual wishes to escape from his or her circumstances but is unable to do so, these ongoing perceptions of blocked escape can evolve into hopelessness. As a result, he or she may lose further motivation to escape or hope for positive changes. Such negative perceptions towards the future are more likely to aggravate the risk of depression.

These findings underscore the differentiation between entrapment and hopelessness as an independent concept. For example, the Cry of Pain model of suicide, within which both entrapment and hopelessness are regarded as prominent predictors of suicidal ideation, has been criticized for its lack of differentiation between these two variables [4,34]. However, some studies have clearly differentiated between entrapment and hopelessness; specifically, unlike entrapment, hopelessness is characterized by an orientation toward the future and the absence of a desire to escape from one’s current circumstances [3,8]. The present results serve as empirical evidence for the notion that entrapment and hopelessness are distinct psychological states by delineating their causal and differential effects on depression.

### 4.2. Moderation Effect of Cognitive Control

As hypothesized, cognitive control showed a negative moderating effect on the relationship between feelings of entrapment and hopelessness. In particular, when the sense of entrapment was weak, there was no significant difference in college students’ hopelessness across the level of cognitive control. However, in the case of having a strong sense of entrapment, hopelessness was much higher in college students with low cognitive control compared to college students with high cognitive control. That is, cognitive control buffers the effects of entrapment on hopelessness, especially among students who struggle with severe entrapment. Similarly, participants with a lower level of cognitive control showed higher overall levels of hopelessness than their counterparts with a higher level of cognitive control, regardless of the level of entrapment.

Such a result highlights that a lack of cognitive control plays an important role in aggravating hopelessness [35,36]. In fact, Abramson and his colleagues (1989), in their hopeless theory of depression, pinpointed that when individuals show cognitive vulnerability that interacts with stressful situations, this increases the likelihood of hopelessness. Cognitively vulnerable individuals tend to make attribution to inner factors and to stable and general causes, expecting negative consequences rather than coping with stressors [37]. As opposed to cognitive vulnerability, cognitive control, the belief that one can eventually control and make differences to stressful situations, enables individuals to reappraise adversity and manage negative emotions over stressors [38]. It is also noteworthy that, supporting the current result, many previous research works on the cognitive vulnerability factor featured in hopelessness theory have used samples of relatively healthy youth and college students [37].

### 4.3. The Moderated Mediation Effect of Cognitive Control

More importantly, cognitive control moderated the mediating effect of hopelessness in the relationship between entrapment and depression. In other words, a greater sense of entrapment can trigger feelings of hopelessness and eventually result in depression, but this mediation effect is significant only among students with lower levels of cognitive control. Thus, if individuals lack cognitive control, they will be more likely to adopt a pessimistic outlook towards the future. Consequently, they may become more vulnerable to depression. In sum, the present findings offer empirical support for the notion that a sense of perceived control over stressors serves as a buffer against the negative mediating effects of hopelessness while one’s psychological state aggravates from entrapment toward depression [8,39].

The current study’s results are consistent with previous study results that focused on the cognitive vulnerability component in the Hopelessness Theory [35]. For example, a meta-analysis conducted by Liu and her colleagues (2015) reported that 23 out of 24 studies indicated that cognitive vulnerability aggravated hopelessness which, in turn, predicted depressive symptoms [36]. These studies particularly found that cognitive vulnerability elicited depression through hopelessness only under ineffective emotional regulation [35,37]. Given that entrapment concerns subjective emotions rather than objective evaluations of circumstances, the present results, consistent with Liu et al. (2015)’s conclusion, suggest that cognitive style (i.e., attribution style and lack of controllability over situations) is crucial when individuals develop hopelessness and, thus, fall into depression [37].

### 4.4. Implications and Limitations

The present findings have meaningful practical implications. First, to prevent depression among college students, it is crucial to assess and alleviate their levels of entrapment. Further, it is important for depressed individuals to be aware of how and why their flight/fight defenses are activated and/or blocked. Open discussions about their perceived reasons for feeling trapped and being unable to cope with their life circumstances are especially critical [40]. For example, Leahy [41] argued that people often remain trapped in unhelpful situations because they have either already invested too much or have too much to lose. Similarly, differentiating between adaptive (e.g., escaping from an abusive relationship) and maladaptive (e.g., suicide as escape) escaping behaviors can also contribute to the disentanglement of such arrested and blocked flight/fight defenses, reducing the risk for depression [8].

Taken together, the present findings underscore the importance of nurturing cognitive control to prevent entrapment from evolving into hopelessness and, eventually, into depression. One example of interventions that aim to enhance cognitive control is mindfulness-based cognitive therapy. Its goal is to help depressed individuals focus on here and now, lessen negative cognitions, and utilize flexible control [42]. In addition, acceptance and commitment therapy (ACT) has also been found to be effective in promoting cognitive control [43]. ACT aims to enhance psychological flexibility and cognitive control by helping individuals disentangle themselves from experiential avoidance and cognitive confusion. Cognitive behavioral therapy, in general, focuses on challenging and modifying clients’ irrational beliefs. In contrast, ACT primarily aims to help individuals fully connect with their present moments [44]. A decrease in experiential avoidance and cognitive confusion contributes to emotional clarification as opposed to entrapment. Consequently, individuals are more likely to avoid further hopelessness and to focus on facing their situations [43].

This study has several limitations, which should be noted. First, the study variables were assessed using retrospective and self-report measures, which are vulnerable to common-method bias. Second, since a cross-sectional research design was adopted in this study, interpretation regarding the causal relationships among entrapment, hopelessness, and depression was limited. Future studies should adopt longitudinal designs and control for possible confounders to ascertain the precise associations between entrapment, hopelessness, and depressive symptoms and delineate the underlying mechanisms. Third, since data were collected from the students of colleges in Seoul and the south-eastern regions of South Korea, generalizations of the present findings should be rendered with caution. Lastly, although the present study conducted analyses by including control variables such as gender and age, future research needs to consider various other confounders (i.e., student physical health, family relations, school performance, etc.). These variables might influence students’ emotional and cognitive functions.

## 5. Conclusions

The current study examined complex relationships among entrapment, hopelessness, and cognitive control in predicting depressive symptoms in college students. More specifically, entrapment was reconfirmed as a primary risk factor for increased depression. In particular, the sense of hopelessness mediates the link between entrapment and depression. These findings enlighten the psychological mechanism whereby individuals feeling like their current situations are inescapable are more likely to envision pessimistic view of their future, and thus are more likely to fall into depression. Such findings also indicate that entrapment and hopelessness need to be differentiated as distinct psychological states. Evaluating levels of entrapment and hopelessness are required in order to successfully help young adults struggling with depression.

With cognitive control, the negative effects of entrapment on hopelessness are found to be alleviated. More importantly, cognitive control buffers the mediating effects of hopelessness between entrapment and depression. When individuals develop belief in overcoming challenges, they are more likely to break their vicious circle of intensifying entrapment into hopelessness, which is protective of further depression. These findings provide empirical evidence for the importance of achieving cognitive control in order to prevent depression in young adults. Interventions designed for depressive young adults to build up control over their challenging circumstances need to be developed. A literature review shows that adaptive escaping such as stopping self-destructive behaviors or relationships, cognitive acceptance of the situation just the way it is, and open discussions about feeling trapped/hopeless can all contribute to achieve cognitive control. Research should also attempt to identify additional factors that can promote cognitive control.

## Figures and Tables

**Figure 1 healthcare-11-01065-f001:**
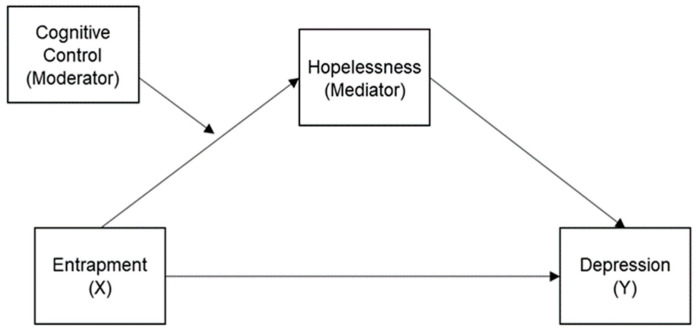
The hypothesized moderated mediation model.

**Figure 2 healthcare-11-01065-f002:**
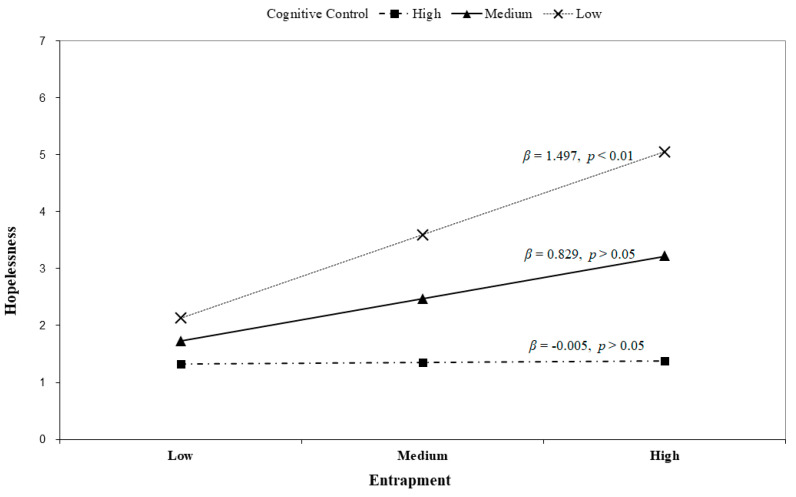
Interaction of cognitive control and entrapment on hopelessness.

**Figure 3 healthcare-11-01065-f003:**
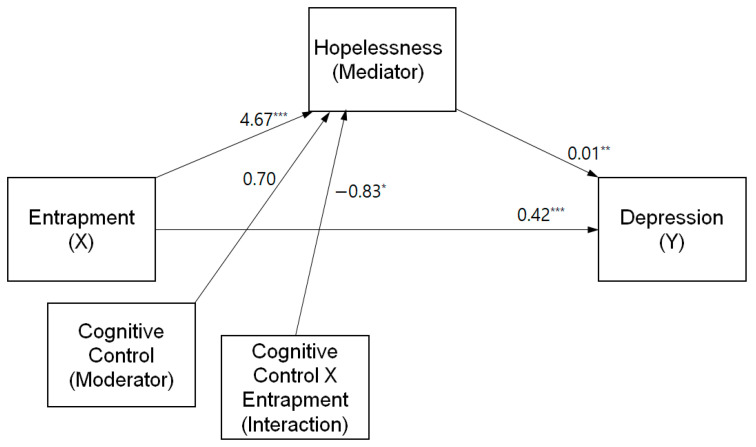
The final model. * *p* < 0.05, ** *p* < 0.01, *** *p* < 0.001.

**Table 1 healthcare-11-01065-t001:** Descriptive statistics and intercorrelations among the study variables (*N* = 367).

Variables	1	2	3	4	5	6
^1^ Entrapment	1					
^2^ Cognitive Control	−0.62 ***	1				
^3^ Hopelessness	0.28 ***	−0.27 ***	1			
^4^ Depression	0.77 ***	−0.58 ***	0.32 ***	1		
^5^ Age	−0.12 *	0.13 *	0.04	−0.10	1	
^6^ Sex (female = 1)	0.33 ***	−0.27 ***	0.03	0.29 ***	−0.47 ***	1
Mean	2.25	4.63	5.60	17.07	21.84	0.51
Standard Deviation	0.90	0.96	5.71	10.34	2.16	0.50

Note: ^1^ = Entrapment, ^2^ = Cognitive Control, ^3^ = Hopelessness, ^4^ = Depression, ^5^ = Age, ^6^ = Sex, * *p* < 0.05, *** *p* < 0.001.

**Table 2 healthcare-11-01065-t002:** Results of mediation analysis (*N* = 367).

Predictors	Dependent Variable: Hopelessness (*R*^2^ = 0.08, *p* < 0.001)			Dependent Variable: Depression (*R*^2^ = 0.61, *p* < 0.001)		
	B	SE	*t*	B	SE	*t*
Entrapment	1.76 ***	0.34	5.12	0.42 ***	0.02	19.75
Hopelessness				0.01 **	0.01	3.44
Age	0.16	0.15	1.03	−0.01	0.01	−0.04
Sex (female = 1)	−0.33	0.70	−0.46	0.06	0.04	1.39
Constant	−1.68	3.58	−0.47	−0.16	0.21	−0.78

Note: SE = Standard Error ** *p* < 0.01, *** *p* < 0.001.

**Table 3 healthcare-11-01065-t003:** Results of moderated mediation analysis (N = 367).

Predictors	Dependent Variable: Hopelessness (*R*^2^ = 0.10, *p* < 0.001)						Dependent Variable: Depression (*R*^2^ = 0.61, *p* < 0.001)					
	B		SE		*t*		B		SE		*t*	
Entrapment	4.67 **		1.47		3.16		0.42 ***		0.02		19.68	
Cognitive Control	0.70		0.79		0.88							
Entrapment × Cognitive Control	−0.83 *		0.33		−2.53							
Hopelessness							0.01 **		0.01		3.29	
Age	1.34		0.15		0.89		−0.01		0.01		−0.06	
Sex (female = 1)	−0.48		0.69		−0.70		0.06		0.04		1.36	
Constant	−2.63		5.06		−0.52		−0.16		0.22		−0.76	
Cognitive Control	Conditional Indirect Effect Entrapment → Hopelessness → Depression											
	B			Boot. SE			95% Boot. CI					
							Lower Limit			Upper Limit		
*M*—1 *SD* (3.8)	0.016			0.007			0.004			0.032		
*M* (4.6)	0.009			0.006			−0.001			0.022		
*M +* 1 *SD* (5.6)	0.000			0.006			−0.014			0.013		

Notes: SE = Standard Error, *M* = Mean, *SD* = Standard Deviation, Boot. SE = Bootstrap Standard Error, 95% Boot. CI = 95% Bootstrap Confidence Interval * *p* < 0.05, ** *p* < 0.01, *** *p* < 0.001.

## Data Availability

Please contact the corresponding author.

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
