# Peer review of "Entrapment, Hopelessness, and Cognitive Control: A Moderated Mediation Model of Depression"

_healthcare, 2023, doi:10.3390/healthcare11081065_

Round 1
Reviewer 1 Report
1, Is it possible that cognitive control moderate the relationship between hopelessness and depression?
2. Need more evidence that hopelessness mediates the relationship between entrapment and depression.
3, Need more information about the sampling, is the sample students a representative sample?
4, It will be better to add information about the control variables?
5, Is there any gender difference on sample students?
6, Need more heterogeneity analysis on school performance, family background...
Author Response
Responses to Reviewer 1 comments
We deeply appreciate your considerate comments. We revised and refined our manuscript by following your advice point by point. Please find the attached file of responses
- It is possible that cognitive control moderates the relationship between hopelessness and depression?
Cognitive control is one of the sub-factors of cognitive flexibility, and refers to one’s tendency to perceive difficult situations as controllable (Heo Shim-yang, 2011). Hopelessness is a pessimistic view of the “future”, the belief that one can hardly do anything to change unhappiness or suffering, and nothing could be done (White, 1989). Conceptually, by examining the relationship between the two variables, we can reasonably anticipate that lack of cognitive control may contribute to the increase in hopelessness. The hopelessness theory of depression especially supports this argument (Liu et al., 2015). According to the HT, the relationship between hopelessness and depression is explained as very close, and it is argued that there are antecedent factors (depression-inducing attributions) that lead to feelings of hopelessness. That is, by making internal, stable, and pervasive attributions to failure, hopelessness can be caused. On the other hand, cognitive control is more about attributing the causes of failure to internal, unstable, and specific, such as efforts for controlling problems (problem-solving). Based on this theory, it can be inferred that cognitive control may serve as effective moderating factor when hopelessness is about to occur.
The Hopelessness Theory of Depression: A Quarter Century in Review. Liu, R., Kleiman, E., Nestor, B., Cheek, S (2015). Clinical Psychology, 22(4), 345-365
- Need more evidence that hopelessness mediates the relationship between entrapment and depression.
We could not find direct supporting evidence for the entrapment--> hopelessness--> depression other than our research result. We believe the value of our research stems from such new trial. However, our research model was not developed without a literature review or theoretical backup. For example, previous research confirmed the conceptual distinction between entrapment and hopelessness, ensuring that entrapment elicits hopelessness (Littlewood et al., 2016). Littlewood and her colleagues tested the multiple indirect mediation model, where the path defeat-->entrapment-->hopelessness-->suicidal behaviors was confirmed. While we examined depression viable rather than suicidal behaviors, depression is a foremost key predictor of suicidal behaviors. Literature review confirms both entrapment and hopelessness are key predictors of depression. And theoretically and conceptually, entrapment precedes hopelessness, we believe our research model was well developed and supported by our literature review.
Littlewood et al., (2016). Nightmares and Suicide in Posttraumatic Stress Disorder: The Mediating Role of Defeat, Entrapment, and Hopelessness. Journal of clinical sleep medicine 12(03).
- Need more information about the sampling. Is the sample students a representative sample?
Our study originally aims to examine general population who registered classes at their university. Although our study collected data on depression, hopelessness, and entrapment, it still targeted general students on campus. We admit the limitations and clearly described in the limitation section that our participants cannot be purely representative of Korean University students. However, research studies using convenience sampling are different from those using national dataset. At least, our data was collected from 3 different universities. Accordingly, number of data was large enough so that it allowed statically meaningful analyses.
Sampling occurred at three universities where both authors teach the classes. We explained details about our study and encouraged students to participate “voluntarily” with a small amount of reward as well as completion of individual consent form. We particularly took care about students’ private information. That’s why we collected participants’ background information to the minimum extent, only including grade level, sex, and ages.
- It will be better to add information about the control variables?
We originally collected data on gender, age, grade level, and academic achievement for demographic information. Admitting the reviewer’s advice, we additionally conducted t-test on the above confounding variables and found that only gender contributed to statistically significant differences. We refined literature review of gender as potential confounder variable in the introduction part (on p.3), as well as, inserted t-test results in the section of preliminary analysis (on p.5-6). Further, we conducted both mediation analysis and moderate mediation analysis by including gender, age, and grade level as control variables. Newly updated results were shown in the result section. In terms of academic achievement, we were not able to conduct proper analysis because there was a large number of missing data.
- Is there any gender differences on sample students?
T-test result shows gender differences on entrapment, cognitive control, and depression. The table was also inserted in the preliminary analysis section (on p. 6).
|
male (n=177) |
female (n=187) |
t |
p |
|||
|
M |
SD |
M |
SD |
|||
|
Entrapment |
1.96 |
.81 |
2.54 |
.89 |
-6.59*** |
.000 |
|
Cognitive Control |
4.89 |
.90 |
4.37 |
.95 |
5.44*** |
.000 |
|
Hopelessness |
5.42 |
6.14 |
5.77 |
5.30 |
-0.58 |
.562 |
|
Depression |
0.69 |
.46 |
1.00 |
.52 |
-5.98*** |
.000 |
However, we could not find any gender differences in terms of mediation and moderating mediation effects. Original mediation and moderating mediation effects did not change. Slight numeric changes after adjusting for confounding variables (age, grade level, gender) were updated in red on p.7.
- Need more heterogeneity analysis on school performance, family background
Unfortunately, we did not collect data on family background. We collected data on academic achievement, but about one fourth of students did not report their academic achievement. Other variables such as gender, age, and grade level, we re-conducted t-test analysis and did not find any difference except for gender. We ensured the needs for heterogeneity analysis for the future research.

Reviewer 2 Report
This research project is very valuable. The manuscript is well written. But the discussion needs a fundamental review. The explanation of the results is not well stated. Besides, the references should be updated.
This research seeks to know how entrapment is placed in depression. This is important and can help therapists. This can attract the attention of therapists and researchers and provide a better understanding of therapeutic strategies.This is a major issue. Elements such as hopelessness and helplessness are well-known and providing new elements and perspectives is helpful. It is well written. The structure of the manuscript is very good. But when I got to the discussion, I found them weak. It did not convince me. It is better for the authors to review the new papers. After updating the discussion and analyzing the components of this research, it is recommended for printing. They should search the new documentation and add new references and use them to explain their results.
Since the discussion in the manuscript is important, I consider it a major revision.
Author Response
Responses to Reviewer 2 comments
Discussion needs to be revised and refined. And new documents and research need to be examined.
We deeply appreciate for your considerate comments. We revised and refined the section of discussion by following your advice. Please find the attached file of responses point by point
We revised and refined discussion sections overall. Changes in discussion were written in red (on p.8~10).
We added specific arguments for each result (1. Mediation effect of hopelessness, 2. Moderation effect of cognitive control in the relationship between entrapment and hopelessness, and 3. Moderated mediation effect of cognitive control over mediation effect of hopelessness)
We also enhanced implication for practice, as well as, limitations of current research (i.e., the research needs to include more various control variables) (on p.11).
Finally, we updated references with more recent studies. (on p. 13).

Author Response
Responses to Reviewer 3 comments
We deeply appreciate for your considerate comments. We refined our manuscript by following your advice. Please find the attached file of responses point by point.
- There is a sentence repetition in sentences in line number 97-102
We appreciate for pinpointing our mistakes. We deleted the repeated part.
- The sentence in line no. 102 stating “this hypothesis is based on the premise that …” needs referencing.
We cited the article written by Littlewood and her colleagues (2016) and put the citation. Littlewood and her colleagues, in their article entitled “Nightmares and Suicide in Posttraumatic Stress Disorder: The Mediating Role of Defeat, Entrapment, and Hopelessness.” examined multiple mediation model of nightmare, defeat, entrapment, hopelessness, and suicidal behaviors. Although their model was not exactly same as ours, we got the insights from their arguments and research results. Littlewood and her colleagues confirmed the multiple indirect mediation model, where the path defeatàentrapmentà hopelessnessà suicidal behaviors was tested. While we examined depression rather than suicidal behaviors, depression is a foremost key predictor of suicidal behaviors. Literature review also ensured that both entrapment and hopelessness are key predictors of depression (Liu et al., 2015; Tylor et al., 2011). Furthermore, theoretically and conceptually, entrapment precedes hopelessness. Hence, we believe our research model was developed and supported by our literature review and theoretical background.
Littlewood et al., (2016). Nightmares and Suicide in Posttraumatic Stress Disorder: The Mediating Role of Defeat, Entrapment, and Hopelessness. Journal of clinical sleep medicine 12(03).
Liu, R., Kleiman, E., Nestor, B., Cheek, S (2015). The Hopelessness Theory of Depression: A Quarter Century in Review.. Clinical Psychology, 22(4), 345-365
Taylor, P., Gooding P., Wood A, & Tarrier N (2011). The role of defeat and entrapment in depression, anxiety, and suicide. Psychol Bull. 137(3), 391-420.
The usage of the word “will” in the Hypothesis 1,2, & 3 should be avoided. Instead the phrases such as “hopelessness mediates” and “cognitive control moderates” can be used.
We deleted the word “will” and changed the whole sentences on p. 3 as follows:
“Based on the literature review and our discussion on previous research results, we formulated three hypotheses. First, we hypothesized that hopelessness mediates the relationship between entrapment and depression. Second, we hypothesized that cognitive control moderates the relationship between entrapment and hopelessness. Lastly, we hypothesized cognitive control moderates the mediating effect of hopelessness on the relationship between entrapment and depression.”
- The details about the ethical approvals and consents i.e. “procedure” section needs to be provided following the “participants” section
On page 4~5 we described the details about data collection process and completion of informed consents under the section of 2.3. Procedures. Details are as follows:
2.3. Procedure
The researchers contacted the faculty members who were teaching undergraduate courses at four-year universities in Seoul and the Southeastern regions of South Korea. Along with faculty members’ permission, students were informed about the study and its purpose through in-class presentations. Participants were provided with informed consent forms and questionnaires. All the respondents were treated in accordance with the code of ethics and conduct of the Korean Psychological Society [29]. In total, 392 survey packets were distributed; 367 participants returned completed informed consent forms and questionnaires (response rate: 94%). Researchers’ contact information, including email addresses and phone numbers, were provided to the participants so that they could contact in order to ask any further question about the research process.
- It is not clear from the paper about how many students met the criteria for clinical depression. Feeling momentarily depressed and clinically depressed could be different things with different etiologies
CES-D suggests the scores between 16~21 indicate moderate risk of depression, while over 21 total points on the scale designates that respondent is at high risk of depression. Our analysis reveals about half of participants (N=174 out of 367, 47.4%) reported moderate to high level of depression. The rest of participants (N=193, 52.6%) were below the score 16. Our study originally aims to examine general population who registered and having classes at their university. Although our study examined such variables as depression, hopelessness, and entrapment, it still targeted general students on campus. Our research results provide implications for how to help and intervene university students who struggle with mild to high risks of depression on campus.
- At the psychological level, cognitive control and entrapment are strongly correlated. In a sense, extent of entrapment could be a result of lack of cognitive control. The moderation effect arises due to the strong correlation between entrapment and cognitive control. In this case an alternate model could be tested. Hypothesis 4: Entrapment mediates effects of cognitive control on hopelessness. This would lead to multi-mediation in series, where entrapment and hopelessness are mediators for effect of lack of cognitive control on depression. The result of this hypothesis can be tested and compared with the Hypothesis 2 and the plausibility of either of these hypotheses can be discussed.
As the reviewers pointed out, the correlation between cognitive control and entrapment was -0.62, showing a strong negative correlation (Table 1). In addition, we partially agree with the reviewer's opinion in that lack of cognitive control can cause entrapment. There is a possibility that a moderation effect appeared due to the strong correlation between cognitive control and entrapment.
However, it is not within the purpose of this study to determine whether cognitive control is a cause of entrapment or not. This study aims to reveal how cognitive control intervenes (buffers) as a moderator in the process of entrapment developing into depression through hopelessness.
Cognitive control is one of the sub-factors of cognitive flexibility, and refers to the tendency to perceive difficult situations as controllable (Heo Shim-yang, 2011). Entrapment refers to a state in which a person has a strong desire for controlling to escape from a given situation, yet negatively evaluates his or her controllability and perceives himself/herself being bound by the current situation (Brown, Harris, & Hepworth, 1995). In contrast, hopelessness is a negative belief and lack of positivity about the future. To be specific, hopelessness indicates the belief that no one can do anything to “change” unhappiness or suffering, and nothing “will” be done (White, 1989). More specifically, hopelessness refers to a state in which the motivation to get out of the situation is gradually reduced in the process of repeated frustration in the process of trying to solve a problem, but entrapment is an emotion that appears when the motivation to control the situation is still high and the recognition that there is no escape is possible.
That is, cognitive control does not necessarily predicts entrapment. Even among individuals possessing cognitive control, entrapment can be temporarily elevated under special circumstances, which can lead to hopelessness, if not properly addressed and sustained. Rather, if cognitive control, as a moderator, is strengthened through interventions such as counseling with individuals who feel trapped, the probability of entrapment leading to hopelessness can be reduced, as well, as risks of developing depression can be reduced. This study was designed to verify whether cognitive control can buffer the mediation effects of hopelessness aggravating into depression by interacting with entrapment.
Heo Shim-yang (2011). The role of cognitive flexibility in the relationship between perfectionism and psychological maladjustment. Seoul National University master's thesis.
Brown, G. W., Harris, T. O., & Hepworth, C. (1995). Loss, humiliation and entrapment among women developing depression: A patient and non-patient comparison. Psychological Medicine, 25(1), 7–21.
Retrieved from https://doi.org/10.1017/S003329170002804X
White, J. L. (1989). The troubled adolescent. New York: Pergamon Press.
- One of the main methodological limitations of this analysis is the lack of controlling for confounders. It is not clear whether any of the confounders were controlled for during the regression analysis. Please discuss what confounders were used. In case, confounders such as age, sex, smoking, BMI, relationship status etc. can be used as primary confounders if the information is collected dufing data collection and check the validity of the results. Since all the variables (hopelessness, cognitive control, entrapment and depression) are correlated adjusting for confounders becomes essential to delineate any partial effects.
We originally collected data on gender, age, grade level, and academic achievement for demographic information. Admitting the reviewer’s advice, we additionally conducted t-test on the above confounding variables and found that only gender contributed to statistically significant differences (except for academic achievement since there was a large number of missing data). We refined literature review of gender and age as potential confounder variable in the introduction part (on p.3), as well as, inserted t-test results in the section of preliminary analysis (on p. 5-6). Further, we conducted both mediation analysis and moderate mediation analysis by including gender as a control variable. Newly updated results (slight numeric changes) were shown in the result section (on p. 6-7).
T-test result shows gender differences on entrapment, cognitive control, and depression. The table was also inserted in the preliminary analysis section (on p.6).
|
male (n=177) |
female (n=187) |
T |
p |
|||
|
M |
SD |
M |
SD |
|||
|
Entrapment |
1.96 |
.81 |
2.54 |
.89 |
-6.59*** |
.000 |
|
Cognitive Control |
4.89 |
.90 |
4.37 |
.95 |
5.44*** |
.000 |
|
Hopelessness |
5.42 |
6.14 |
5.77 |
5.30 |
-0.58 |
.562 |
|
Depression |
0.69 |
.46 |
1.00 |
.52 |
-5.98*** |
.000 |
However, we could not find any gender difference in terms of mediation and moderating mediation effects. Original mediation and moderating mediation effects did not change. Slight numeric changes after adjusting for confounding variables (age, grade level, gender) were updated in red on p.7.

Round 2
Reviewer 1 Report
1, Table 3 is hard to understand. It is better to report the results in a figure like figure 1.
2, It is better to control more demographic and social and economic characteristics of sample students in the analysis.
Author Response
Reviewer 1 (Round 2)
We deeply appreciate for your considerate comments. We refined our manuscript by following your advice.

1. Table 3 is hard to understand. It is better to report the results in a figure like figure 1.
We changed figure 1 slightly by renaming the variables. For example, we changed “Mo” into “Moderation” so that readers can recognize the research model more clearly.
We also put new figure on pp.8 . Figure 3 combined the stastical results from table 3 and our original research model (Fig. 1). As you advised we believe the figure 3 can summarize our results better in order to help readers understand.
2. It is better to control more demographic and social and economical
charactersitscs of sample students in the analysis.
As reported earlier version of reponse, we originally collected students' demographic data on gender, age, grade level, and academic achievement NOT including socio-economic status.
Last time (Round 1 revision), We admitted the reviewer’s advice and additionally conducted t-test on the above control variables and found that only gender contributed to statistically significant differences. We refined literature review of gender and age as potential confounder variable in the introduction part (on p.3), as well as, inserted t-test results in the section of preliminary analysis (on p. 5-6). Further, we conducted both mediation analysis and moderate mediation analysis by including gender as a control variable. Newly updated results (slight numeric changes) were shown in the result section (on p. 6-7)
The following T- test result shows gender differences on entrapment, cognitive control, and depression. The table was also inserted in the preliminary analysis section (on p.6).
|
|
male (n=177) |
female (n=187) |
T |
p |
||
|
M |
SD |
M |
SD |
|||
|
Entrapment |
1.96 |
.81 |
2.54 |
.89 |
-6.59*** |
.000 |
|
Cognitive Control |
4.89 |
.90 |
4.37 |
.95 |
5.44*** |
.000 |
|
Hopelessness |
5.42 |
6.14 |
5.77 |
5.30 |
-0.58 |
.562 |
|
Depression |
0.69 |
.46 |
1.00 |
.52 |
-5.98*** |
.000 |
However, we could not find any gender difference in terms of mediation and moderating mediation effects. Original mediation and moderating mediation effects did not change. Slight numeric changes after adjusting for confounding variables (age, grade level, gender) were updated in red on p.7.
We additionally mentioned the limitations of not including enough demographic information in the research implication sections on p.11, as follows.
“Lastly, although the present study conducted analyses by including control variables such as gender and age, future research needs to consider more various confounders (i.e., student physical health, family relations, and school performance, etc.). These variables might influence students’ emotional and cognitive functions.”
